# Integrated Geophysics and Geomatics Surveys in the Valley of the Kings

**DOI:** 10.3390/s20061552

**Published:** 2020-03-11

**Authors:** Francesco Porcelli, Luigi Sambuelli, Cesare Comina, Antonia Spanò, Andrea Lingua, Alessio Calantropio, Gianluca Catanzariti, Filiberto Chiabrando, Federico Fischanger, Paolo Maschio, Ahmed Ellaithy, Giulia Airoldi, Valeria De Ruvo

**Affiliations:** 1Department of Applied Science and Technology, Polytechnic University of Turin, 10129 Turin, Italy; 2Department of Environment, Infrastructures and Territory Engineering, Polytechnic University of Turin, 10129 Turin, Italy; 3Department of Earth Science, University of Turin, 10124 Turin, Italy; 4Department of Architecture and Design, Polytechnic University of Turin, 10125 Turin, Italy; 53DGeoimaging, 10125 Turin, Italy; 6Geostudi Astier, 57123 Livorno, Italy

**Keywords:** Valley of the Kings, geomatics, geophysics, archaeology, laser scanning, close-range photogrammetry, simultaneous localization and mapping, electrical resistivity tomography, ground-penetrating radar, geomagnetic survey

## Abstract

Recent results within the framework of the collaborative project The Complete Geophysical Survey of the Valley of the Kings (VOK) (Luxor, Egypt) are reported in this article. In October 2018, a team of geomatics and geophysics researchers coordinated by the Polytechnic University of Turin worked side by side in the VOK. Topographic measurements in support of geophysical surveys and the achievement of a very large-scale 3D map of the Eastern VOK were the two main objectives of the geomatics campaign. Innovative 3D metric technologies and methods, based on terrestrial laser scanning (both static and mobile) and close-range photogrammetry were employed by the Geomatics team. The geophysical campaign focused on the acquisition of Electrical Resistivity Tomography (ERT), Ground Penetrating Radar (GPR) and high spatial density Geomagnetic (GM) data. ERT new data around KV62, both inverted in 2D sections and added to the previous ones to perform a new global 3D inversion, confirm the previous results showing both conductive and resistive anomalies that have to be explained. GPR timeslices showed some interesting features in the area in front of the KV2 entrance where GM gradient map also presents localized anomalies. In the area SSW of the KV2 the GM gradient maps evidenced also a large semicircular anomaly which, up to now, has no explanation. The potentialities of using magnetic techniques as a complement to other non-invasive techniques in the search for structures of archeological significance have been explored. The application of modern and innovative methods of 3D metric survey enabled to achieve a complete 3D mapping of what is currently visible in the valley. The integration of 2D/3D mapping data concerning visible elements and hypothetical anomalies, together with the recovering in the same global reference system of underground documentation pertaining to the Theban Mapping Project, prefigure the enhancement of multi-temporal site representation. This strategy enables the fruition development of the already discovered archaeological heritage, using modern criteria of valorization and conservation.

## 1. Introduction

Nowadays, it is globally accepted that reconstructing the complete history of a cultural site, including structures of archaeological significance yet to be discovered and encompassing the sphere of immaterial cultural values, is the necessary approach for the proper preservation and valorization of Cultural Heritage (CH). Essential elements of this approach include extensive sharing of new technologies, developments and results within the scientific community and outreach operations aimed at the general public, as recognized for several years [1].

In recent years, the debate on the cultural, social and economic value of promoting built heritage with the support of digitalization and digital systems has increased and developed in many directions. With a certain degree of schematization, it is possible to assert that the issues that at the international level arouse the greatest interests are the need for 4D recording (3D plus time), including multidisciplinary information [2], of different sources, formats and contents, and their sharing, distribution and dissemination to the users by the world-wide-web using ad hoc developed platforms and/or applications. A second broad theme is that of 3D inventories and related historical information, which is necessarily connected to the need to use standards. The last one is more technically linked to the themes of visualization and communication in a user-friendly way, allowing different possible users to manage and query the data, selecting useful information according to their interests, using shared digital inventories as well as virtual or augmented reality visualization techniques.

The present research is focused on analytical skills related to the space dimension of individual techniques and the integration of different methods: geophysical investigations in connection to survey techniques. The goal is to take advantage of the broader integrated multidisciplinary comparison with their interaction and collaboration [3,4].

The area of our investigations is the Valley of the Kings (VOK) in Luxor, Egypt, arguably the most important necropolis of Ancient Egypt. The VOK has always captured the attention and the imagination of Egyptologists and scholars as well as the general public. Most famous of all is the royal tomb of the golden young king Tutankhamun (code name KV62), who ruled Egypt from 1333 to 1323 B.C., i.e., by the end of the 18th New Kingdom Dynasty. This tomb was discovered with a nearly intact funerary treasure by Howard Carter in 1922 [5]. In 2015, the egyptologist Nicholas Reeves proposed a theory [6] according to which Tutankhamun’s tomb may be part of a larger tomb belonging to Queen Nefertiti. Reeves’ hypothesis was based on a close examination of high-resolution 3D laser scan photos taken by the Factum Arte organization to create a replica of KV62 [7]. To test this theory, a ground-penetrating radar (GPR) scan from the inside of KV62 was authorized by the Egyptian Ministry of Antiquities (MA) and performed by Watanabe in November 2015. This preliminary GPR scan seemed to confirm Reeves’ hypothesis. The finding was called “the discovery of the century” and was reported by news media around the world [8,9]. However, a second GPR scan performed in 2016 by a National Geographic team [10] could not confirm the initial finding by Watanabe. In 2017, our team was selected by the MA to carry out a third, conclusive scan, carried out in February 2018, the results of which were published in [11]. Unfortunately, our work showed that Reeves’ hypothesis is not supported by our GPR scans carried out from the inside of KV62.

The third KV62 GPR scan was conducted within the framework of a broader project, also authorized by the MA and called “The Complete Geophysical Survey of the Valley of the Kings.” According to this project, our team was given the privilege of carrying outdoor geophysical and geomatic surveys in any place of potential interest within the VOK. This is indeed a very challenging task, which indeed has just started and will take several years to complete. The objective of the present article is to present some of the preliminary results of potential interest and, more importantly, the strategy and the rationale for integrating geomatics and geophysical work. We shall focus, in particular, on results obtained in the Eastern branch of the VOK, where most pharaonic tombs are located. We did carry out investigations also in the Western Valley (also called the Valley of the Monkeys), where only two important tombs are located (the tombs of Ay and that of Amenhotep III), and where, in 2009, a team of Egyptian archeologists led by Zahi Hawass discovered foundation deposits and other intriguing archeological finds [12], suggesting that an until-now undiscovered pharaonic tomb is still lying hidden below the sands of the Western Valley. Preliminary results on our geophysical surveys in the Western Valley were reported in [13], but the search for the undiscovered pharaonic tomb is still going on.

Returning to our work in the Eastern Valley, we decided to conduct integrated Geomatics and Geophysical work according to the following strategy.

The Geomatics group performed activities aimed at two main objectives. The first objective was topographic measurements in support of geophysics surveys, i.e., providing specialist support for the georeferencing the data collected during the geophysics carried out in parallel within the framework of the project “The Complete Geophysical Survey of the Valley of the Kings (VOK)”. The second objective was the achievement of a very large-scale map of the Eastern Valley of the Kings using innovative metric 3D technologies and methods, based on static and mobile (SLAM based) laser scanning, with the integration of close-range photogrammetry, to update and improve the data of a previously existing documentation project [14,15,16].

Concerning the Geophysical surveys, additional ERT profiles were performed mostly in the area around KV62 (the tomb of Tutankhamun), in order to integrate and clarify the results performed by the authors in the same area in May 2017. ERT surveys have indeed already proven to be effective in the VOK environment [13,17]. Furthermore, high-density Geomagnetic (GM) surveys have been executed on some selected areas where GPR surveys were carried out previously by the same team. Both GPR [18] and GM [19,20,21] surveys have already proven to be effective in the context of ancient Egypt archaeology and they are usually used in combination to take advantage of the comparison among the two results [22,23,24]. With respect to the investigations reported in this paper, the areas investigated by GM surveys were a subset of the area in front of KV2 entrance (the tomb of Ramses IV) and subsets of the areas near KV10, KV11, KV9 and KV56.

To the best of our knowledge, this was the first time that an integrated 3D metric survey, such as topography, photogrammetry, Light Detection and Ranging (LiDAR), and an Mobile Mapping System (MMS) based on Simultaneous Localization and Mapping (SLAM) was carried out in the VOK, as well it was the first time that a high-density GM surveys were performed in the surroundings of the tomb of Ramses IV.

## 2. The Geomatics Surveys: Acquisition and Processing

The aim of the survey campaign accomplished by the geomatics team was the accurate 3D documentation of the eastern VOK, with one of the main goals being a very detailed surface model able to support the geophysics analysis and surveys, and all the other needs related to the 3D mapping of the valley.

The technologies were, as briefly reported before the LiDAR (in-ground applications, also known as Terrestrial Laser Scanning (TLS)) and an innovative MMS based on SLAM algorithms (Zeb-Revo RT by Geoslam), that offers a very fast acquisition phase. The main goal is to generate a multi-sensor 3D model of the valley, obtained by the data integration derived from different survey methods and techniques, mainly classified as range-based techniques (Lidar and MMS) and image-based methods (close-range photogrammetry). The following sections describe briefly the methods applied and the instruments employed to fulfill the stated purposes (Table 1).

The present section has been organized with the aim of presenting the measurement and processing of metric data, starting from the topographic networks on which all the detailed metric surveys are based followed by a general description of all the other employed techniques. Briefly, the terrestrial laser cloud processing and the innovative ortho-photo generation followed by digital photogrammetry technologies will be presented, together with the results of the mobile mapping system scans, acquired by the afore-mentioned portable system ZEB-Revo RT.

### 2.1. Topographic Survey for Plano-Altimetric Referencing

The first 3D metric survey started by using GNSS (Global Navigation Satellite System) techniques, in order to define a complete network using the international standard UTM WGS 84 reference system. For defining the UTM WGS 84 reference system during the GNSS acquisition two vertices (VK3 and VK2) were measured for more than 7 h in order to use the PPP (Precise Point Positioning) approach [25] and obtain the coordinate of the vertices in the UTM system. Furthermore, after the standard acquisition of the other baselines, the GNSS network were also optimized by using traditional survey methods (by means of a total station) in order to provide more accurate point heights. These activities were carried out using a robotic total station (Image station) by Leica Geosystem [26]. In particular, the new topographical network was built using the vertices already existing on the site and constituting the Theban Mapping Project (TMP), to perform calculations and adjustments in the international UTM WGS 84 system and to have a suitable plano-altimetric referencing (Figure 1). In fact, the aim of this phase was to update the TMP topographic network, which is available online, and therefore globally accessible, although in a local coordinate system. This task will offer the chance to organize a 3D Geographic Information System (GIS) project that can be planned with the aim to obtain a complete multidisciplinary data sharing of the available documents (both cartographic and alphanumeric) that could be developed using open-source software. It can also be helpful and necessary for architectural, archaeological and environmental survey operations, as well as to obtain the reference of all the measurements and the related models to a single Cartesian system. This procedure made it possible to take into account the error propagation and consequently to guarantee the required accuracies, both for laser scanning and terrestrial photogrammetry surveys.

#### Data Processing

The data processing of the network was performed in three different steps. First of all, due to the lack of accessing a permanent GNSS network, the PPP approach was employed in order to evaluate the coordinates of the aforementioned two vertices using the software developed by the Natural Resources Canada’s Canadian Geodetic Survey [27], that allows to compute high accuracy positions of raw Global Navigation Satellite System (GNSS) data. Furthermore, starting from those points, all the base-lines of the network were computed using Leica Geo Office [28] and exported in order to be used for the final adjustment of the network, that has been performed in MicroSurvey STAR*NET [29]; the choice of using this software for the final adjustment was followed since in this software it is possible to combine in a very easy way different data (GNSS and topographic measurement) for computing the network. After the adjustment procedure, the vertices adjusted coordinates (UTM WGS 84 36 N) and the related accuracies were evaluated (Table 2).

The topographic measures were useful for supporting the electrical resistivity tomography and magnetometry surveys detailed in the following sections. The survey was, in fact, essential in order to ensure the spatial relation of the new surveys with the ones of the previous campaign. Moreover, another important aspect was to connect the geophysical results to the land morphology and to the known positions of the existing tombs using at first the local reference coordinate systems (developed by the TMP) and then moving the whole survey to a global reference system (WGS84-UTM). The first operation involved tracking on the ground the profiles of the ERT investigations related to the last measurement campaign, in order to design and implement the new surveys, extending the survey coverage area and avoiding the mapping of already investigated areas. The magnetic survey areas had the perimeter measured by means of a total station in order to localize the magnetic data with respect to the graves and to the morphology of the terrain.

### 2.2. Laser Scanning Survey

Upon consideration of the extension of the area, the required quality of the scans and the wide measurement range (maximum 330 m of distance even in full sun), the phase shift laser FARO Focus X330 by CAM2 was chosen as TLS system (Table 3). The aim was to perform an accurate 3D survey, offering a very detailed surface model able to support the geophysics prosecutions and all other needs related to the 3D mapping of the valley.

Thus, in order to cover the whole area, 108 scan positions (at a density of 6 mm at 10 m) were placed in the general path of visits, excluding the interior of the tombs, on the top of the hills and along the border of the cliffs. The laser scanner is equipped with an integrated digital camera that allows us to acquire the images necessary to associate the color to each acquired point. In order to connect the scans to the reference system of the GNSS network, 76 ground control points (GCPs) were measured by a total station. The GCPs were then employed for georeferencing the resulting point cloud that is obtained by merging together the entire set of single scans.

#### Data Processing

After acquiring data with the traditional TLS systems, the software CAM2 SCENE by FARO [30] was used for the 3D processing and documentation. The adopted approach consisted of the cloud-to-cloud registration, which uses the well-known Iterative Closest Point (ICP) algorithms [31] to connect with each other the largest number of scans. The aim is to have a faithful reproduction of the archaeological site. Moreover, a second data registration based on the previously surveyed set of topographic GCPs was performed in order to georeference the survey in the UTM WGS 84 system. After the registration process, it is possible to observe a residual error of 2 cm. The registration steps required a scrupulous work, as each of the scans was not in the sightline of all the others: for this reason, the registration process was performed by clustering the different scans and registering apart numerous groups and subgroups, in order to contain the propagation of the registration error (Figure 2). In this way, it was possible to align almost the totality of the scans (102/108).

The TLS system allows us to associate the metric component (point cloud) with the radiometric component (images) acquired with the integrated digital coaxial camera, assigning an RGB value to each point. The obtained clouds were exported and subsequently imported into the CloudCompare [32], in order to merge them in a single one (102 scans with a total weight of 40 GB), and a geometric filtering process(one point each 1 cm) was performed (Figure 3).

### 2.3. Mobile Mapping System Survey

The innovative MMS (Mobile Mapping System) technology based on SLAM (Simultaneous Localization and Mapping) algorithm and an Inertial Measurement Unit (IMU) was chosen in order to extend and evaluate the contribution of the multisensory approach, offering a very quick acquiring phase. The first pioneering solution was made available [33]. In this study, a handheld 3D scanner called Zeb Revo RT by GeoSlam (Table 4) was used for the survey of some of the paths across the Valley outside the tombs. The complete 3D model was generated by more than 30 MMS scans (Figure 4), which lasted from a few minutes to twenty minutes per scans and acquired more than 120 million of points distributed quite homogeneously around the surfaces covered by the mapping trajectory execution. The mean density of the point cloud was around 8000 pt/m^2^. The outdoor context was very challenging for the mapping system, but the SLAM-based clouds usefully integrated areas that were poorly covered or under-sampled by the TLS fixed scans.

#### Data Processing

The SLAM algorithm implemented in the technology solved the problem of positioning by recognition of similar geometries in the continuously extracted LiDAR profiles from the instrument rotating head.

Some common problems of drift errors in the trajectories and non-closed loops in the final point clouds occurred in these complex contexts of acquisition execution (Figure 4). In the post-processing phase, as many studies show [34,35], it was possible to re-process and correct these typical problems. Since this range-based system, which has high performance in indoor environments, is neither equipped with devices able to determine the spatial location of the scans, nor with a GNSS receiver, the problem of positioning was solved by recognition of similar geometries in the continuously extracted clouds to the fixed LiDAR point model. The alignment of complete ZEB scans to the LiDAR DSM of the tombs area provided mean deviation errors ranging between 5 and 50 cm (Figure 5).

Most of the time, when 3D metric survey products with high-quality radiometric contents are needed (such as detailed and accurate orthophotos), photogrammetric applications are integrated with MMS and LiDAR data. Because the MMS clouds are not colored, and the RGB component of the LiDAR cloud is derived from images acquired at a medium quality, a photogrammetric survey was performed, as discussed in the next section.

### 2.4. Photogrammetric Survey

Photogrammetry has proven, in the last decade, to be a reliable and cost-effective survey technique presenting undoubtedly advantages with respect to other metric survey methodologies [36]. Because of the impossibility of performing a UAV photogrammetric survey in the archaeological site of the VOK, a terrestrial survey was conducted instead. Many hundreds of high-resolution images were acquired using two different DSLR (digital single-lens reflex) cameras and the usual digital photogrammetry approach with widely overlapping images was adopted. At the end of the camera calibration and image orientation step, 382 images on a total of 403 were successfully aligned (Sony ILCE-7RM3-CMOS 43.6 Mpx—sensor size 36 mm × 24 mm—Image size 7952 × 5304 pix—focal length 24 mm). However, more than 3000 images were acquired for documentation purposes (Canon EOS 5DS R-24–105mm f/4 24–105 mm), but only a part of these images were used for the photogrammetric process, representing the same object at a fixed focal length and with the correct acquisition geometry.

The GCPs used for LiDAR data are the same used to process and orient the images. The photogrammetric survey was also applied to collecting images in the general paths of visits, on the top of the hills and from the borderline of the cliffs (Figure 6).

#### Data Processing

The photogrammetric block orientation was realized by software Agisoft Metashape [37], which combines image matching and point cloud generation through SfM (Structure from Motion) algorithms; the metric products represented in Figure 7 were therefore generated. Details about the quality of the survey and the accuracy of the georeferentiation are presented in the following Table 5.

In a perspective of sensor integration and data fusion, the images oriented at the end of the BBA (Bundle Block Adjustment) were used for enhancing the radiometric quality of the TLS point cloud; this was performed via the option “Colorize Dense Cloud” of Agisoft Metashape, and was possible because the LiDAR cloud was georeferenced in the same coordinate system of the photogrammetric block. 

## 3. The Geophysics Surveys: Acquisition and Processing

The geophysical activities within this campaign was focused on extending the ERT surveys made during the past missions and on integrating the information obtained with the Ground Penetrating Radar (GPR) by high spatial density Geomagnetic Surveys (GM). In Figure 8, a map of all the geophysical surveys executed within the Project is reported, evidencing the new acquisitions in terms of ERT and GM surveys.

### 3.1. ERT

The Electrical Resistivity Tomography (ERT) surveys within the VOK were executed in different periods of time and focused along two main areas of interest: above and near the hill hosting the Tutankhamun KV62 tomb and near the KV2 and KV5 tombs, within and along the sides of the main tourist path, from the rest area toward the entrance of the Valley (Figure 8). For all the acquisitions, a 72 channel resistivity-meter (IRIS SyscalPro Switch) was used with different electrode spacings (in the 0.5 to 2 m range) and dispositions, depending on the available space for the surveys and topographic constraints (electrodes locations for all the surveys are reported in Figure 8). For the KV62 area, three different campaigns were conducted: the first in February–March 2017 (ERT 1), the second in May 2017 (ERT 8-13 and 26-27) and the third in October 2018 (ERT X, Y, W, Z). These latter profiles are indicated in Figure 8 with dashed cyan lines and were acquired with 48 electrodes at 1.5 m spacing. For the KV2 and KV5 area, two ERT 3D datasets (ERT 2-3) and one ERT profile (ERT 4) were acquired in February–March 2017. Multiple electrode arrays, i.e., dipole–dipole, Wenner–Schlumberger, reciprocal Schlumberger and pole-dipole (including also, when possible, cross cable measurements), were the configurations adopted for the surveys to combine potential resolution and depth of penetration. A robotic total station was used to determine the exact positions of electrodes and to survey the topography of the study area. More details on the data acquisition for the KV62 area are reported in [2].

#### Data Processing

Data processing and inversion were performed using ERTLab64™ 3D software, developed by Multi-Phase Technologies and Geostudi Astier, which uses a Finite Elements (FEM) approach to model the subsoil by adopting mesh of hexahedrons to correctly incorporate terrain topography. The inversion procedure is based on a least-squares smoothness constrained approach [38]. Throughout the inversion iterations, the effect of non-Gaussian noise is appropriately managed using a robust data weighting algorithm [38,39]. Before inversion, a series of filtering procedures were adopted, aimed at removing noisy and/or unreliable data. The VOK is indeed a very challenging site in terms of contact resistances at electrodes, which often are in the order of hundreds of kOhm: this occurs particularly in the areas occupied by debris, where even large amounts of salt-water and clay around electrodes was not effective in reducing contact resistances. Therefore, measurements with instrumental standard deviation > 5%, either negative or > 50,000 Ohm·m apparent resistivity values and recognizable outliers were removed from the datasets before inversion. On the remaining data, different processing was performed. All the survey lines acquired in the different periods of time in the KV62 area were combined in a unique 3D model, visualized and interpreted also in relation to geomatic data acquisitions. The remaining surveys were inverted independently. For all datasets, several iterations were necessary to achieve the convergence of the inversion process starting from a homogeneous model representing the high resistive VOK environment (5000 Ohm·m homogeneous medium). The noise control through the standard deviation reweighting algorithm on the outliers and the remodulation of the roughness factor were crucial in ensuring a final model robust enough against possible artifacts.

### 3.2. GPR

GPR data were collected with an IDS^®^ Stream-X 200 MHz GPR unit (integrating 7 dipoles with vertical polarization) and positioned with an automatic Total Station Leica TS-15 (Figure 8). The whole GPR system comprises the shielded antenna box accommodated within a 4-wheels cart equipped with a magnetic encoder triggering the system, the K2FastWave digital acquisition device, a laptop control unit and a 12 Volt battery unit. Seven parallel radargrams with a mutual spacing of 12 cm were simultaneously collected within each swath (width = 72 cm). Acquisition parameters consisted of 512 samples/scan within a time-window of 160ns. Several swaths of the system were collected across two adjacent areas beside KV2 tomb’s entrance and along the facing visitors’ path (Figure 8). As a result, GPR data cover a total area of 2.715 m^2^, corresponding to 34.661 m of GPR scans and 1.449 profiles.

#### Data Processing

GPR data were post-processed with GPR-SLICE software (www.gpr-survey.com). In order to remove background noise and to enhance signals of interest, the following filters were applied: (1) static correction, (2) background removal, (3) band-pass+gain, (4) spectral-whitening, and (5) Hilbert transform (Figure 9).

A ground radar-wave velocity of 0.115 m/ns was estimated with a hyperbola fitting method. According to this velocity, a time window of about 50 ns corresponds to a reliable investigation depth around 3 m below ground level.

Hilbert transformed data were used to directly compile GPR pulse volumes for each dataset with variable cell size, depending on the spacing among the profiles and corresponding to half of the lines spacing. Cell interpolation was then applied to these volumes along *X* and *Y* directions in order to fill the gaps between adjacent lines. Topographic correction of the data was performed by warping the GPR volume over a digital terrain model generated from the topographic data recorded by the total station. Horizontal time-slices together with time-slices parallel to the topographic surfaces and vertical cross-sections were then extracted from the interpolated volumes. Finally, the main reflections from variable depth ranges were projected onto the horizontal plane and synthesized as individual 2D maps (time-slices overlay analysis).

### 3.3. GM

The magnetic surveys were carried out along parallel profiles within the selected areas (Figure 8). The results presented in the following concerns only the subsets of the area in front of KV2, respectively MAG-3 and MAG-2 in Figure 8. Two different magnetometers were used for the acquisitions: a GEM System GSM-19GW which is an Overhauser Proton Magnetometer [40,41,42] and GEOMETRICS MAFM Laser Pumped Cesium Magnetometer [43,44]. The surveys were carried out according to the acquisition parameters shown in Table 6. Both magnetometers measured in every point along the profiles (with a sampling interval according to acquisition parameters), the modulus of the Total Earth Magnetic Field (TMF) at two different heights above the ground. In this way, in every reading point, the Vertical Gradient of the TMF (VGTMF) can be estimated too. Indeed, it is well known, from the theory, that the vertical gradients of TMF are particularly sensitive to anomalies caused by superficial bodies [45,46,47].

The corner points of each area were gathered with a total station. The actual extension of the surveyed areas is reported in Table 7.

#### Data Processing

Data processing was carried out using MATLAB© [48] and Surfer© [49]. Magnetic data acquired in each area, because of their abundance and the relatively short time taken to gather an area underwent a gross statistical analysis in order to give proxies of the TMF in the VOK. The results of this analysis are summarized in Table 8.

The first step of GM data processing was mainly aimed at distributing the readings along with each measuring profile and to give a raw-data interpolation. 

As far as the GEM is concerned, the mean and the standard deviation of the intervals between two consecutive readings were respectively: 0.2 and 0.02 m. It is possible to evaluate that, without losing meaningful information, an interpolation interval along the profile directions equal to 0.25 m could be effective. This choice has also the advantage of generating an isotropic grid (0.25 m × 0.25 m) which facilitates any further processing operation. The higher sampling rate of the MAFM instrument (Table 8) conversely resulted in reduced measuring intervals among the readings. This has the effect of increasing the data abundance but also introduce some noise. To attenuate the effect of the 50 Hz noise the MAFAM raw data were re-sampled at 50 Hz thus obtaining intervals between two consecutive readings variable from 0.02 to 0.04 m. Finally, with the same criteria used for GEM data, the TMF and VGTMF values were recalculated on an isotropic 0.25 m × 0.25 m grid. Further processing steps applied to the resampled GEM and MAFM maps were as follows. 

De-striping: striping is nearly always present in magnetic data. It consists of a striped aspect of the maps mainly due to the back and forth acquisition paths. To reduce the striping effect it is usually enough to subtract to each profile its mean value. This procedure is less effective when in the area local high-intensity anomalies are present, like those produced by causative bodies such as pipes, power cables, manholes and so on. These high-intensity anomalies, in fact, alter the means of the profiles they belong to, thus enhancing the striping. In the data this situation was particularly present in MAG-3 area where the de-striping, performed subtracting the mean of the TMF along the profiles, was not fully effective.

De-shifting: shifting is essentially due to the non-constant velocity of the operator and it consists of a zigzag or “stepped-line” aspects of any maxima alignment across the profiles. It is particularly evident on strong aligned anomalies as, for example, those produced by underground cables or pipes. Sometimes cross-correlating the profiles allows us to find the shifts which give the best alignment of maxima. It may also happen, especially when using fast acquisition rates, that the shifts are small and a generic 2D smoothing filter produces a satisfying result in reducing shifting effects.

Further processing was a wavelength high-pass (HP) spatial filtering [50]. HP filtering allow to reduce the amplitudes of the small size anomalies, related to more surficial sources, and enhance the ones of the larger anomalies due to deeper causative bodies.

## 4. Results

### 4.1. Geomatics

The multi-sensors 3D data acquisition, range-based and image-based, led to the targeted result of obtaining a 3D multi-content and multiscale 3D map of the VOK, the completeness of which is limited to the surface of the ground, to the tourist routes, to the steep cliff, but excluding the all graves in the area. As planned, the complete Digital Terrain Model constitutes the spatial base to compare the terrain surface characteristics with the underground tombs (which position and morphology is derived from the TMP), and obviously with the anomalies found by the geophysical analyses.

This result, together with the already mentioned localization using the global UTMWGS84 system, opens the possibility for multiple different uses and benefits, some of which are highlighted in the following.

The first undoubted advantage is to correlate the anomalies deriving from geophysical investigations to the reality of the terrain, managing these data in a single reference system. This comparison that relates known buried assets, the reality of the factual situation, and the location of anomalies deriving from the possible presence of other buried and not yet discovered structures, could be a prelude to a possible planning of targeted checks by means of excavation tests that validate the effectiveness of the integrated methods.

The availability of the accurate point cloud allows to derive orthophotos with the same resolution of the sampling level of the cloud, i.e., 1 cm, which is configured as a high-quality product on which to base the management of the site in a broad sense (visitor flow, maintenance, cleaning and safety management) (Figure 10).

The integrated use and managing of different survey products in the GIS environment is also very interesting as it allows to represent the external environment, mapped according to the standards of numerical cartography, and the presence of underground tombs derived from the Theban mapping project in a joint visualization (Figure 11). The huge scale of the map and the upgraded mapping of modern artifacts, together with the availability of being able to interrogate the DSM to know the height at any point, would allow, for example, the possibility of planning any visit plan by the disabled public. As a case in point, Figure 12 highlights the present arrangements of the entrance of KV11 e KV55 tombs.

Finally, an application based on the models so far obtained by the 2018 3D survey campaign was prepared to offer a virtual visit through 3D HOP (3D Heritage Online Presenter), an open-source software package for the creation of interactive web presentations of high-resolution 3D models, oriented to the Cultural Heritage field [51] (Figure 13). 3DHOP allowed the creation of an integrated interactive visualization of the photogrammetric, LiDAR and SLAM based 3D models so far generated, embedding all of them directly inside a standard web page based on HTML and JavaScript components. Thanks to the integrated multi-resolution 3D model management, the user can visualize the models from a personal computer without the use of specific software or hardware. There is the possibility to perform a virtual tour of a set of four tombs (only the surrounding of the external area) and thanks to the hotspot function it is possible to obtain relevant information on the inspected tomb’s entrances (Figure 13). The skin (fonts and colors) of the webpage developed in the framework of this research is inspired to the one of a live demo presenting a sculpture of the head of Tutankhamun, accessible on the software webpage [52].

### 4.2. Geophysics

#### 4.2.1. ERT

All the ERT surveys performed depict an underground environment constituted by (Figure 14): massive bedrock (around 3000–5000 Ohm·m) with rare conductive bodies (<3000 Ohm·m) and local high resistivity anomalies (>10,000 Ohm·m). Low resistivity values are commonly associated to either fine-grained material (e.g., debris infill), or water flow through permeable layers and/or along discontinuities. High resistivities characterize voids and/or cavities, including, for instance, open joints in fracture zones.

From the results obtained through the inversions of all the ERT datasets, it is possible to evidence the correct identification of known voids and structures in the VOK environment, demonstrating the effectiveness of the chosen interpretation approach. High resistive anomalies are indeed present in the data when the surveys cut, or are near to, known underground tombs, see for example the results reported in Figure 14 (bottom left) near the KV5 tomb, and in the KV62 area. The 3D dataset acquired around the mound of Tutankhamun KV62 tomb is particularly effective in defining the known presence of the tomb and in locating other potential interesting anomalies. With respect to already available interpretations of data from previous survey seasons in the KV62 area [2], the merging of all the surveys performed in the different time periods (see Figure 8) allowed a better model definition and a more precise definition of position and shape of evidenced anomalies, which turn out to be more contained in space and less extended (Figure 14—comparison between top and mid- left results). 

In the KV2 survey area, the electric resistivity variation reconstructed from ERT 2 and 3 surveys are also reported in Figure 14 (right side). These were obtained by processing according to a 3D approach the combined dataset of the different 2D survey lines acquired in the areas (see Figure 8). Both surveys depict the presence of localized low resistivity anomalies (blue colors) due to either important lithological discontinuities (gypsum/shale levels intercalated into the Theban Limestone), or presence of debris material, perhaps derived from repeated flash flood events, infilling natural or anthropic voids, or recent excavations subsequently back-filled. More extensive knowledge on the past archeological activities in the area is essential to better understand the nature of these anomalies. Particularly interesting is the relevant low resistivity area evidenced in the ERT-3 area. Indeed the shape and orientation of this area compare well with one of the already discovered KV5 rooms. This anomaly can be therefore potentially associated with KV5 environments full of flooding debris, some of which already known by weeks [17], some still to be totally or partially excavated. Local high resistive anomalies are also present and potentially interpreted as cavities, or intensively fractured zones. These are particularly evident in the E side of the ERT 3 area where a significant high resistivity anomaly is visible and persistent for significative depths.

#### 4.2.2. GPR

In Figure 15, the most significant results from the GPR surveys are summarized and presented. The reflection energy distribution across the site is imaged under the form of the absolute envelope amplitude, with a blue-to-red color scale used to show the whole range of amplitudes detected. Deep blue hues represent parts of the investigated volume with low reflection amplitude values, i.e., where little or no radar signature was detected. Yellow–red colors identify the zones with the highest amplitude values, i.e., the main reflectors.

The most significant results of the radar survey were obtained from the area in front of KV2, as represented in the radar maps and cross-sections in Figure 15. It is particularly notable in this area, between 0–0.6 m and 0.6–1.2 m depth (top of Figure 15), a localized high-amplitude anomaly (yellow circle). The same anomaly is still visible down to the 2.6–3.5 m depth range and, partially, on cross-section (bottom of Figure 15). Such an isolated anomaly within the relatively uniform front yard of KV2 has a non-clear origin and can be associated with a still unknown deposits concentration. More extensive knowledge on the past archeological activities in the area is essential to better understand the nature of this anomaly.

Similarly, strong reflectors with overall significant lateral continuity define an irregularly sloping volume within the 0.6–3.5 m depth range, this reflector is clearly visible in cross-section as a linear feature (bottom of Figure 15). Geometry and continuity at depth of such series of anomalies can be interpreted as the bedrock top surface, dipping away from the tomb’s entrance towards the visitors’ path, which would be consistent with descriptions of the site by Ayrton [53]. If these considerations are correct, the presence of a shallow cover made of natural sediments overlying and leveling a deeper and irregular bedrock morphology (including backfill of old excavation pits) [14,54,55] could be assumed in this area. 

Another strong reflector is the linear feature adjacent and parallel to the visitors’ path clearly visible on the radar map within the 2.6–3.5 m depth range (bedrock/boulders/containment structure in Figure 15) and in cross-section. Given its persistence, geometry and position, it could represent an anthropic perpetual confinement structure (wall or cluster of boulders) to the KV2-A area.

In the area aside the KV2, where also GM surveys were performed (see later), generally weaker reflectors are evidenced, given also the unfavorable ground conditions due to uneven surface that did not allow a uniform survey distribution. As a result, anomalies look smoothed as a consequence of the broad interpolation applied to the data. Between 0 and 1.2 m depth, some reflectors are however noticeable. A slight variation in reflection patterns in the south-eastern margin of the area could point at a local undulation of the bedrock top surface within the 0.6-1.2 m depth range.

Along the sides of the visitors’ path and across the southern portion of this area, a great deal of underground utilities were detected within the first 60 cm from the surface (top of Figure 15). Between 0.6 and 1.2 m depths and deeper, strong reflection patterns cut across the north-westernmost section of the area. By similarity with other features recognized in this depth range in the other investigated areas, it is reasonable to assume this to be the limestone bedrock top surface.

#### 4.2.3. GM

In Figure 16 the VGTMF maps of MAG-2 and MAG-3 areas, after the application of a square 7 × 7 unweighted moving average filter, are presented. Considering the 0.25 × 0.25 grid of the original map, the applied filter attenuates the anomalies with wavelengths shorter than 1.5 m. In Figure 16, MAG-3 map comes from the processing of MFAM gradient data while MAG-2 map from GEM data. Both maps underwent clipping of high-intensity anomalies to recover low-intensity features, which are more likely related to eventual archaeological remains and/or structures.

The MAG-3 VGTMF map is strongly influenced by the many artifacts at its borders: the entrance of the KV2 tomb at W; the shelter for tourists at S; and the N and E containing walls. Some anomalies, however, are visible in the center of the area where the GPR also evidenced some localized reflections. A combined interpretation of these anomalies is therefore possible underlining the eventual presence of backfill of old excavation pits.

The MAG-2 VGTMF map is dominated in its N side by the effect of the shelter. The E side presents anomalies that are likely attributable to some technological facility. Of some interest are a nearly square-shaped anomaly, which can be seen in the NE corner with the sides oriented to SN and EW and may be correlated with the SE corner of the MAG-3 map. Particularly relevant is also the evidence of a semicircular slightly positive anomaly, with a diameter of about 15 m, in the central-southern part of the map.

## 5. Conclusions and Future Developments

The performed analysis and the presented results concerning the geophysics and geomatics survey in the Valley of the Kings are very promising and some perspectives can be highlighted.

From the Geomatics point of view, the representation of the Valley of the Kings that we have obtained is a 3D digital map rich in high-resolution geometric details from which level contours, ortho-photos and other relevant 3D metric data have been derived. This project allowed the evaluation of the relationship between the ancient tombs and the morphology of the land and the landscape, presenting some important updates between the present site situation represented by the 3D survey (2018) and the state-of-the-art survey at the period of the Theban Mapping Project site campaign. The ground-based scans and related 3D models have been successfully integrated with photogrammetric surveys, obtaining metrically controlled high-resolution textures.

Even though the surveys and the relative analysis carried out have been limited to outdoor areas in the VOK (and not at its full extent), this work has been the first step towards a new Atlas of the Valley of the Kings using modern geomatic techniques able to integrate topographic data as well as geophysical and geological information. The final aim will be integrating and complementing the information collected by the TMP, that was published in the ’80 s.

From the Geophysics point of view, additional ERT information and insight is presented in this article concerning the two anomalies that were previously reported at some distance from the KV62 tomb [2]. Interesting anomalies are also evidenced by ERT in the other investigation areas. We think that these are an interesting first sight of the potentiality of the method in the VOK environment. Further surveys with reduced electrodes spacing and denser survey alignments could be foreseen to further increase the knowledge on the evidenced anomalies. More extensive knowledge on the past archeological activities in the area is essential to better understand the nature of these anomalies. The authors have also investigated areas near KV2 using magnetometric equipment, revealing new evidence of anomalies of apparent anthropic origin in that area. To the best of our knowledge, this was the first time that high-density magnetometry was used in the Valley of the Kings. The potentials of this technique in this context is yet to be established, indeed the presence of utilities and other structures complicated data acquisition and interpretation. However, anomalies related to facilities show different wavelengths and shape with respect to the ones referable to shaft or tombs. Therefore, careful survey design and processing could significantly improve the discrimination between what could be considered noise and useful signals. On the whole, it is the Author’s opinion that magnetic techniques can complement other non-invasive, geophysical techniques such as GPR and ERT in the search for structures of archaeological significance in this context too. Particularly interesting is the correspondence of GM anomalies and reflection patterns from GPR surveys in front and around the KV2 area.

The Polytechnic University of Turin collaboration proposes to extend the 3D mapping of the Valley of the Kings to the underground tombs in order to update the metric and thematic documentation of the extraordinary archaeological heritage of the Valley. In this respect, our work would build upon the extraordinary effort carried out previously by the TMP [5,6].

From the Geophysics point of view, the authors’ recommendations for future work are to:Complete the GPR survey of the tourist path in the Eastern Valley of the Kings. So far, the area covered by GPR surveys goes from the Valley entrance gate to KV35; GPR scanning of the areas of the tourist path beyond KV35 to the end of the Eastern Valley and side branches of the Eastern Valley are yet to be carried out;Continue ERT and magnetic investigation of specific areas in the Eastern Valley of the Kings, where the presence of interesting anomalies of possible anthropic origin have been found in the data collected so far.

From the geomatics point of view, the authors recommend to extend this work to the interior of selected tombs of particular interest, with the immediate goal to make possible a virtual tour of these tombs to be offered as a possible experience to disabled people as well as visitors of the Grand Egyptian Museum and other museums of interest in Egypt. 

Extending geomatic work to the 3D mapping of the underground environment of the Valley of the Kings in Luxor could also provide valuable information for the future development of virtual tools, accessible via the web, like those offered by augmented reality. 

## Figures and Tables

**Figure 1 sensors-20-01552-f001:**
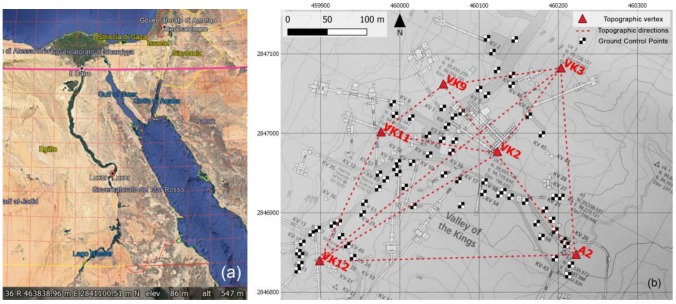
The Valley of Kings (VOK) in Luxor: (**a**) VOK location in UTM (source Google Earth). (**b**) The measured topographic network (red triangles) and the localization of Ground Control Points (black and white markers) derived from total station measurements, superimposed to the Sheet 1/70, KV and WV extracted from the Atlas of the Theban Mapping Project (TMP), page 13, Plan 1:2500, Contour interval 10 m.

**Figure 2 sensors-20-01552-f002:**
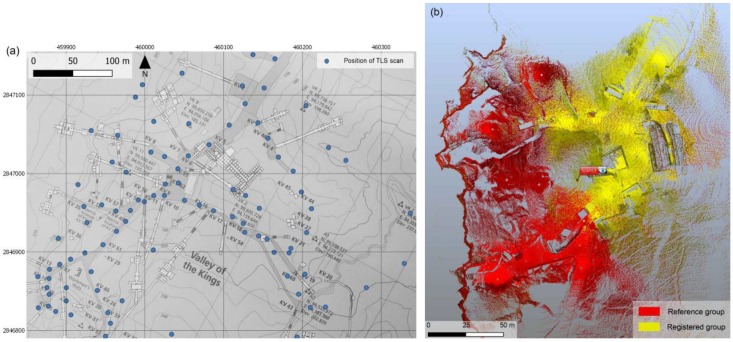
Laser scanning survey: (**a**) position of each of the acquired TLS scans (blue dots), superimposed to the Sheet 1/70, KV and WV extracted from the Atlas of the TMP, page 13, Plan 1:2500, Contour interval 10 m. (**b**) The registration process of the scans groups performed via the software Faro SCENE.

**Figure 3 sensors-20-01552-f003:**
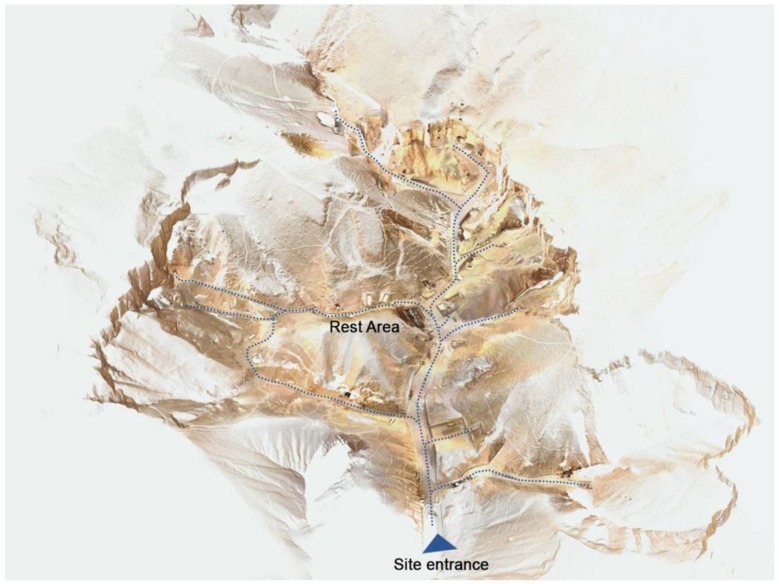
Image of the resulting merged point clouds (102 single scans) with more than 2 billion of points, and with superimposed visiting paths (open and closed).

**Figure 4 sensors-20-01552-f004:**
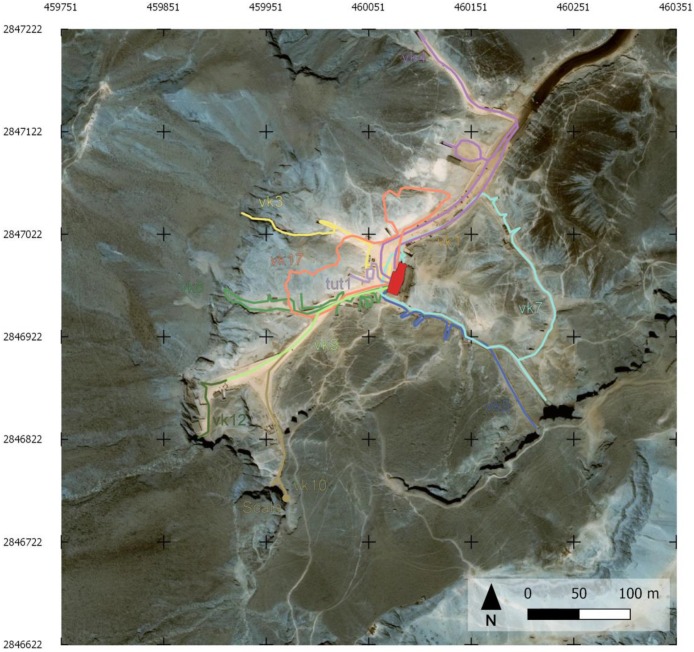
General map of the more than thirty scan trajectories acquired by the MMS platform in the VOK. Different colors represent different surveyed paths.

**Figure 5 sensors-20-01552-f005:**
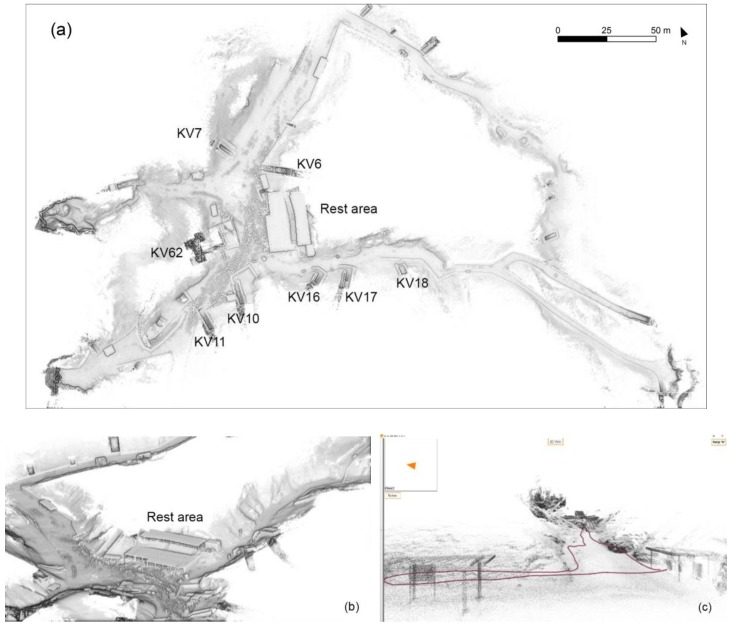
Mobile Mapping System survey: (**a**) the plan view of several registered point clouds acquired by MMS (b,c) Some views of the 3D model, in shaded features (**b**) and with the trajectory (**c**).

**Figure 6 sensors-20-01552-f006:**
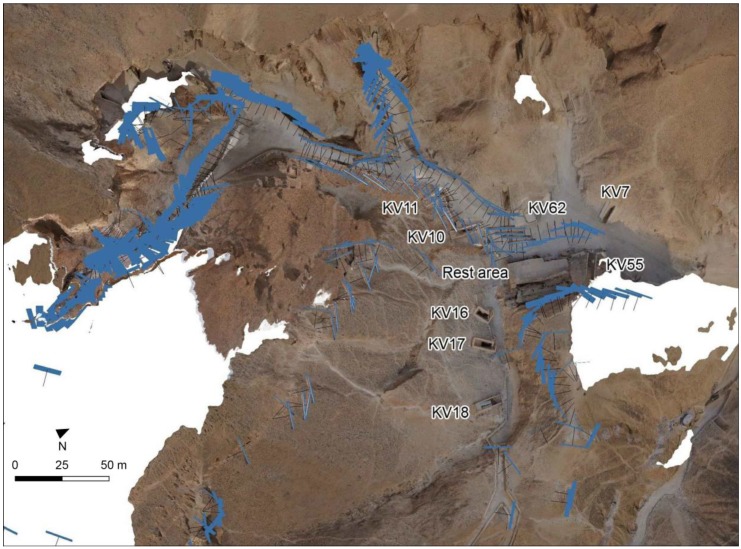
The position of the estimated camera centers after the photogrammetric block orientation, superimposed on the orthophoto generated from the subsequent photogrammetric process.

**Figure 7 sensors-20-01552-f007:**
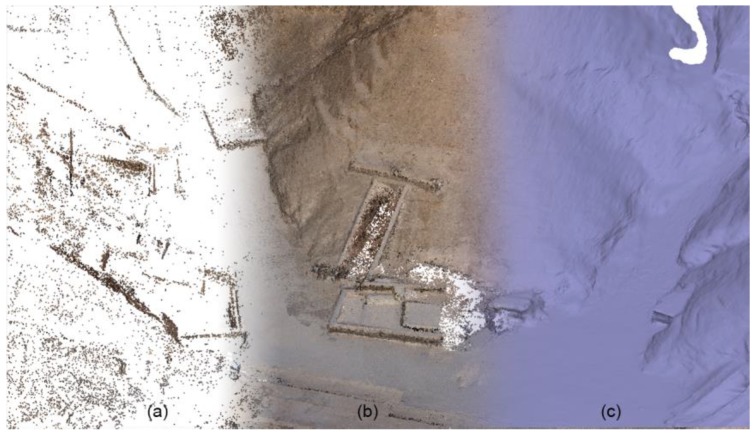
A picture presenting the different 3D photogrammetric products: Sparse point cloud (**a**) dense point cloud (**b**), triangulated mesh (**c**).

**Figure 8 sensors-20-01552-f008:**
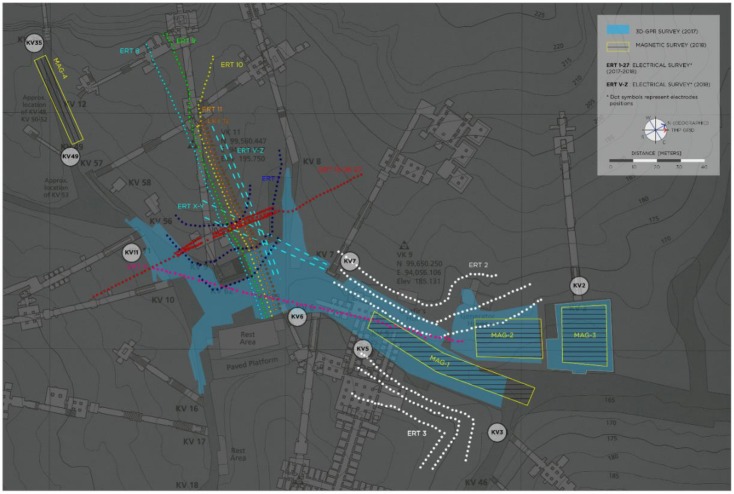
Map of the geophysical surveys performed in the area which contribute to the presented results. The different colored lines refer to the different Electrical Resistivity Tomography (ERT) surveys executed.

**Figure 9 sensors-20-01552-f009:**
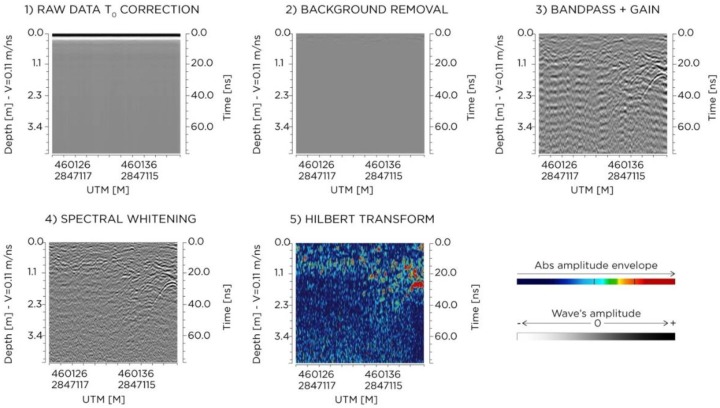
Example of processing step effects on a single radargram.

**Figure 10 sensors-20-01552-f010:**
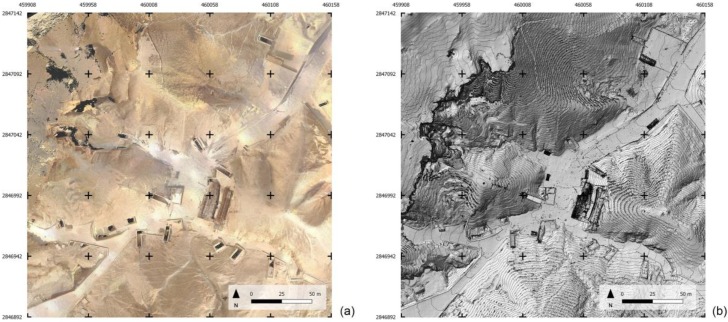
Orthophoto generated from the terrestrial photogrammetry dataset (**a**); Digital Elevation Model (DEM) with contour lines generated from the TLS dataset (**b**).

**Figure 11 sensors-20-01552-f011:**
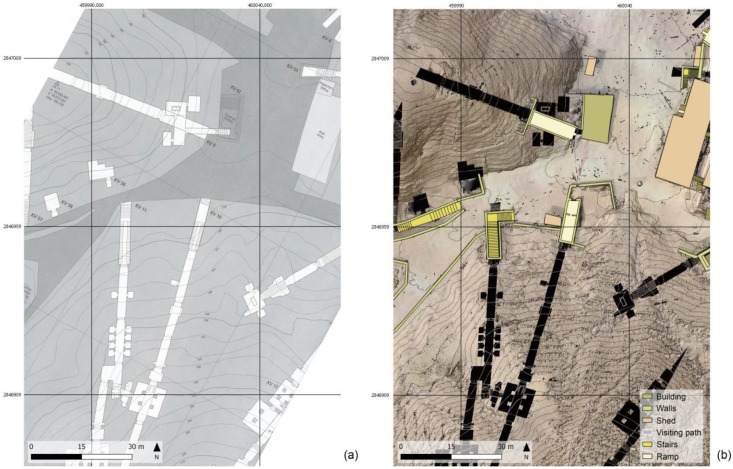
Different survey products and underground tombs derived from the Theban mapping project sheets integrated into the Geographic Information System (GIS) environment. (**a**) Sheet 5/70, KV (3/4) extracted from the Atlas of the TMP, page 21, with superimposed the new global grid (50 m interval), Plan 1:400, Contour interval 2 m; (**b**) the corresponding area with the up-to-date geoinformation in GIS environment, Plan 1:100, contour interval 1 m, same global grid as on the left.

**Figure 12 sensors-20-01552-f012:**
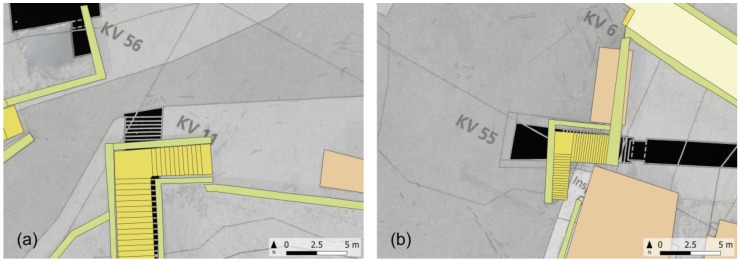
Zooming in from the Figure 11, upgraded mapping of modern artifacts concerning the entrance of KV11 (**a**) and KV55 (**b**) in comparison with the Theban Mapping Project.

**Figure 13 sensors-20-01552-f013:**
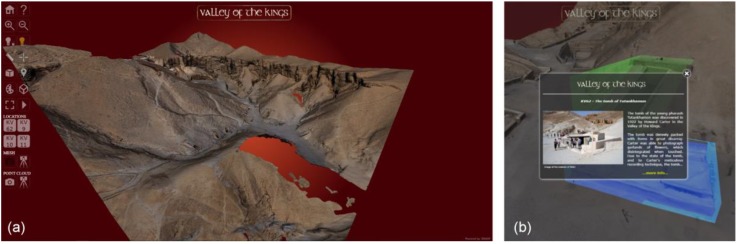
Virtual visit of VOK: (**a**) interactive Web presentations of the high-resolution 3D model; (**b**) hot-Spot function embedded in the 3D viewer.

**Figure 14 sensors-20-01552-f014:**
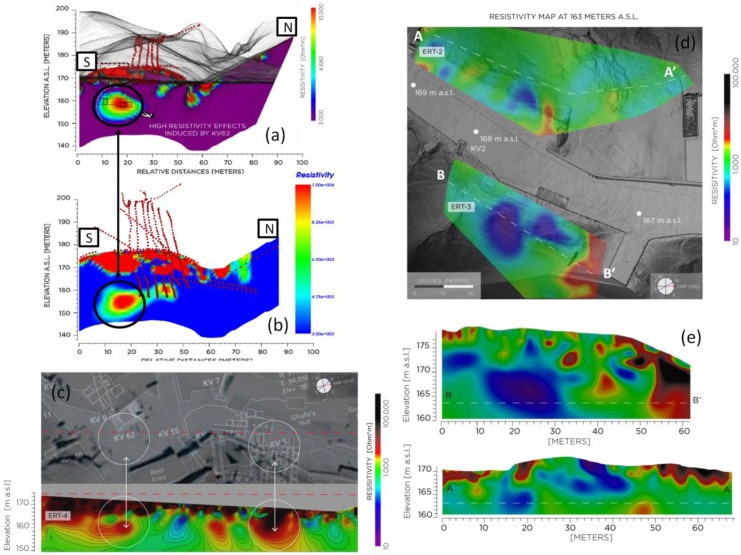
Results from ERT surveys: (**a**) and (**b**) representative cross-section of the 3D model over the KV62 tomb (a from 2017 surveys and b from 2018 surveys); (**c**) the electric resistivity variation reconstructed by ERT-4 and comparison with known voids. The electric resistivity variation reconstructed by ERT-2 and -3 shown by representative maps (**d**) and their associated cross-sections (**e**). Dashed white lines show where each map and its vertical section intersect.

**Figure 15 sensors-20-01552-f015:**
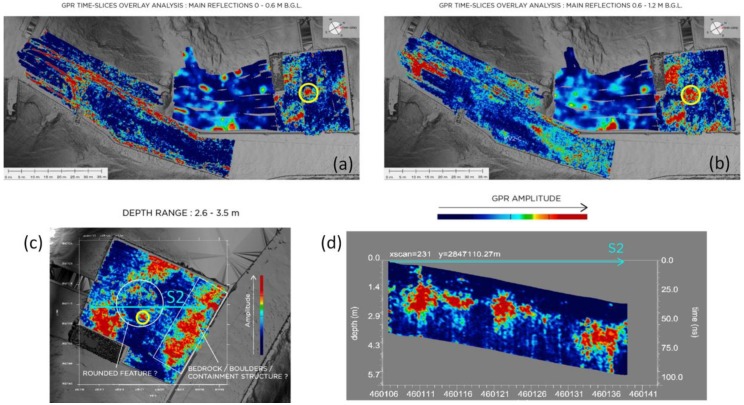
Results of GPR time slice overlay analysis over and around the KV2-A area. In (**a**) and (**b**) a sequence of time slices for two representative depth ranges is reported. In (**c**) and (**d**) a focus on the area in front of KV2 with: a time slice for a representative depth range (c) and the corresponding cross-section S2 (d) showing the distribution at depth of the main anomalies.

**Figure 16 sensors-20-01552-f016:**
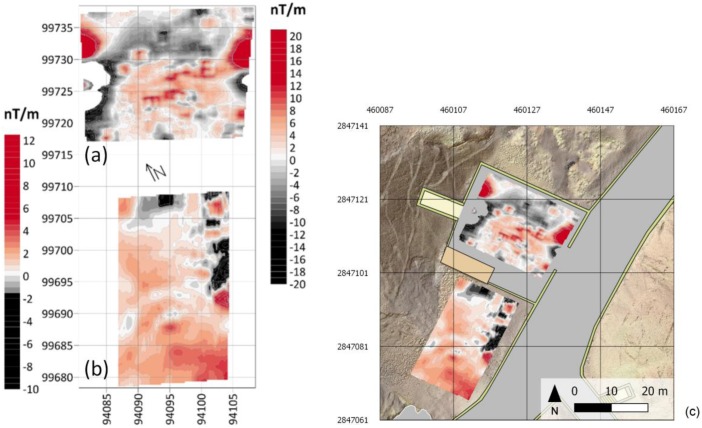
VGTMF map (local coordinates system) of (**a**) MAG-3 and (**b**) MAG-2 areas. The left color scale refers to MAG-2, the right color scale refers to MAG-3. The color scales have different ranges to enhance the peculiarity of the two areas. In (**c**) the same VGTMF maps are integrated in the geomatics GIS environment (georeferenced in the UTM WGS 84 system).

**Table 1 sensors-20-01552-t001:** Geomatics instruments used in the data acquisition phase.

Type of Survey	Systems	Employed Sensors
Range-based	TLS	Faro^®^ Focus3D X330
	MMS	GeoSLAM ZEB Revo
Image-based	DSLR cameras	Canon EOS 5DS R
		Sony ILCE-7RM3
Topography	GNSS	Geomax Zenith 35
	TS	LEICA NOVA M250

**Table 2 sensors-20-01552-t002:** Coordinates of topographical vertcies characterized by the accuracies shown in the last three columns on the right.

Vertex	Est (m)	North (m)	Elevation (m)	RMSE (m)		
				Planimetric	Altimetric	Total
VK3 [fixed]	460,203.524	2,847,082.428	210.444	-	-	-
A2	460,222.492	2,846,847.688	215.056	0.002	0.003	0.003
VK11	459,976.111	2,847,001.856	207.945	0.001	0.002	0.002
VK12	459,898.698	2,846,839.672	213.886	0.002	0.003	0.003
VK2 [fixed]	460,123.102	2,846,977.174	210.864	-	-	-
VK9	460,054.954	2,847,062.238	197.300	0.001	0.002	0.002

**Table 3 sensors-20-01552-t003:** Main specifications of the Faro^®^ Focus3D X330 scanner used for the Terrestrial Laser Scanning (TLS) survey.

Main Specifications of the Faro^®^ Focus3D X330 Scanner	
Operational range	0.6–330 (m)
Ranging error	± 2 (mm)
Vertical/horizontal FoV (Field of View)	300/360 (°)
RGB sensor (size of the final stitched image)	Up to 70 (Mpx)
Acquisition speed	Up to 976.000 (pt/s)

**Table 4 sensors-20-01552-t004:** Main specifications of the GeoSlam ZEB Revo RT platform used for the Mobile Mapping System (MMS) survey.

Specifications of the GeoSlam ZEB Revo RT Scanner	
Wavelength	905 (nm)
Eye-safe laser	Class 1
Laser speed (repetition rate)	100 (Hz)
Laser lines	100 (lines/s)
Scan speed	× 2.5
Maximum range	15–30 (m)
Points density	~ 43,200 (pps)
3D measurement declared accuracy	2–3 cm (relative accuracy)
FoV	270° HFOV/100° VFOV
Weight/portability (head + data logger)	~ 2 (kg)

**Table 5 sensors-20-01552-t005:** Information concerning the quality of the photogrammetric processing, and the achieved accuracy, represented by the RMSE of the markers used for the generation of the photogrammetric model.

Average Distance (m)	Average GSD (mm/pix)	Coverage Area (km^2^)	Tie Points	Projections	Reprojection Error (pix)	Markers’ RMSE (m)
104	8.5	0.04	719,697	2,351,918	0.937 pix	0.015

**Table 6 sensors-20-01552-t006:** Acquisition parameters of the Geomagnetic Surveys (GM) survey.

	GEM	MFAM
TMF/GRAD Sampling rate (S/s)	5	1000
Height of the lower sensor (m)	0.3	0.3
Height of the upper sensor (m)	1.3	1
Profile interval (m)	0.25	0.25

**Table 7 sensors-20-01552-t007:** Details concerning the extension of the MAG-3 and MAG-2 surveyed area.

Site	Dimension (m × m)	Instrument	Area (m^2^)	*n*. of Profiles	Total Profile’s Length (m)
KV2a	≈ 20.5 × 26.5	MFAM	545	83	2199
KV2b	18 × 30	GEM + MFAM	540 (×2)	73 (×2)	2190 (×2)
TOTAL			1625	229	6579

**Table 8 sensors-20-01552-t008:** Summary of the results of the gross statistical analysis performed on magnetic data in MAG-3 and MAG-2.

Area	Date	Begin (U.T.)	End (U.T.)	n. of Samples	Mean (nT)	St. Dev. (nT)	St. Dev. of Mean (nT)
MAG-3MAFM	Mon Oct 22 2018	14:35:47	15:25:46	1765244	4.195565 × 10^4^	1.1234 × 10^2^	8 × 10^−2^
MAG-2 MAFM	Tue Oct 23 2018	04:28:05	05:10:42	1634117	4.199083 × 10^4^	7.86	1 × 10^−2^
MAG-2 GEM	Tue Oct 23 2018	05:44:51	06:47:03	11330	4.198193 × 10^4^	6.89	7 × 10^−2^

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
