# Peer review of "Integrated Geophysics and Geomatics Surveys in the Valley of the Kings"

_sensors, 2020, doi:10.3390/s20061552_

Round 1
Reviewer 1 Report
Dear Authors
Thank you for the presented paper. I think that great work lies behind the acquisition of the data but the presentation of the results is not satisfactory and, in the present form the paper is not publishable.
The paper, indeed, is more a report of the activities realized in the archaeological site that a real integration of methods as declared by the title (and so on…). So, it is necessary that the authors focus their efforts to give more scientific proof to their experience. This is, in particular, mandatory for the paragraphs concerning the geomatics surveys.
The introduction is quite interesting but it should be improved because the references are really limited. I suggest to the authors to use references where ERT, GPR, and MAG are used with success for archaeological studies. Similar considerations for geomatic applications.
As concerns geophysical surveys, the results showed are interesting, in particular for ERT. But their presentation is too summarized and in some parts, impossible to understand. To increase the comprehension of the results, it is necessary for example to give information about the distribution of the electrodes. Figure 14 shows the results of the 3d electrical behavior of the subsoil near the KV62 tomb but no information about the disposition of the electrodes and the configuration adopted is given by the authors.
Further, it is not clear if ERT2 and ERT3 are related to a 3d acquisition or are simple 2d lines. Clarify, please
A lot of electrical and electromagnetic anomalies present in the few GPR and ERT data reported in the paper are not explained and discussed. The authors are invited to elaborate more on these aspects. In my opinion, finally, GM surveys are not useful for archaeological feature detection. The reason is the presence of utilities and other structures and the presentation of the results can be neglected, and maybe the discussion inherent this methodology can be totally avoided. Finally, the quality of the image is too low.
I suggest to the authors to reorganize the paper rewriting the parts of the results and their discussion and then, to re-submit the work.
Reviewer 2 Report
I would like to congratulate the authors on this very interesting study. The paper has a very good structure and all parts are mostly very well described. I highlight especially the parts with the data processing. The topic itself is very attractive to readers. The studied field was imaged by several methods which complemented each other. The English used in this paper is very good.
Nevertheless, I would recommend several minor changes which, I believe, will improve the quality of the paper.
Some typos revision will be needed, e.g., vertices vs vertexes (chose one of them to be more consistent). Using sources might be improved. Examples: You’re talking about new technologies in the introduction and cite a paper from 2004 [3]. You mention recent studies and cite a paper from 2007 [4]. Page 7: ‘…, as many studies show [19]’. If you talk about many studies, it would be good to mention more than one reference. Table 1 is a bit confusing; some lines would improve its readability. In section 2.4 you mention two cameras but you speak only about the Sony one (in Table 1 also Canon was mentioned but it can be reminded again in the text). The usage of a grid in Figure 11 would simplify the comparison of the two images. Some very brief introduction of used methods will be needed, especially for ERT, GPR, and GM. On what principle do they work, what are their advantages and disadvantages, what are their limitations? In the conclusions, some explanation of using these three specific methods might clarify the study. What is the profit of combining the methods? Are the results in accordance? Why yes? Why not?After including these revisions I recommend this paper to be published.
Reviewer 3 Report
See attachment.

Reviewer 4 Report
Paper focused on a geophysical and geomatical surveys performed to the Valley of the Kings (VoK) (Luxor, Egypt). This paper constitutes an interesting integration of the used methodologies.
Results are very interesting therefore I think that paper merit to be published with minor revisions.
Abstract needs to be more concise, focus on location, need for investigation methods used expected outcomes. Introduction: authors should be clear how your work could be advances the state of the art or the knowledge in this field. I think that the Authors could be focused this paragraph to specify the peculiar aspects of novelty of this paper with respect to other studies performed from other authors. I believe that would be much more informative if Authors provide additional details about the survey strategy and the processing procedure.Suggest following break down:
a) raw data initial interpretation b) Modelling description and outcomes c) processing steps used and why? d) final interpretation of processed resultsAuthor Response
Please see the attachment
